# Unifying Discrete and Continuous Representations for Unsupervised Paraphrase Generation

**Mingfeng Xue**♠ **Dayiheng Liu*** **Wenqiang Lei**♠ **Jie Fu** **Jian Lan**♠
**Mei Li** **Baosong Yang** **Jun Xie** **Yidan Zhang**♠ **Dezhong Peng**♠ **Jiancheng Lv**♠
♠ College of Computer Science, Sichuan University
♠ Engineering Research Center of Machine Learning and Industry Intelligence
mingfengxue@stu.scu.edu.cn    losinuris@gmail.com

## Abstract

Unsupervised paraphrase generation is a challenging task that benefits a variety of downstream NLP applications. Current unsupervised methods for paraphrase generation typically employ round-trip translation or denoising, which require translation corpus and result in paraphrases overly similar to the original sentences in surface structure. Most of these methods lack explicit control over the similarity between the original and generated sentences, and the entities are also less correctly kept. To obviate the reliance on translation data and prompt greater variations in surface structure, we propose a self-supervised pseudo-data construction method that generates diverse pseudo-paraphrases in distinct surface structures for a given sentence. To control the similarity and generate accurate entities, we propose an unsupervised paraphrasing model that encodes the sentence meaning and the entities with discrete and continuous variables, respectively. The similarity can be controlled by sampling discrete variables and the entities are kept substantially accurate due to the specific modeling of entities using continuous variables. Experimental results on two benchmark datasets demonstrate the advantages of our pseudo-data construction method compared to round-trip translation, and the superiority of our paraphrasing model over the state-of-the-art unsupervised methods.

## 1 Introduction

The task of paraphrase generation aims to reconstruct a sentence in a different surface structure (grammar and words) while preserving its underlying semantics. It plays an essential role in many downstream tasks like text summarization (Cao et al., 2017; Zhao et al., 2018), machine translation (Zhou et al., 2019; Thompson and Post, 2020), question answering (Dong et al.,

2017; Buck et al., 2018; Zhang et al., 2022), document modeling (Zhang et al., 2021), and others. Supervised paraphrase generation methods require large-scale, manually annotated paraphrase datasets, which are labor-intensive to build. This makes unsupervised paraphrasing an emerging research direction.

Existing unsupervised paraphrase generation methods primarily employ round-trip translation (Mallinson et al., 2017; Wieting and Gimpel, 2018; Guo et al., 2021) or denoising (Hegde and Patil, 2020; Guo et al., 2021; Niu et al., 2021), which leads to four flaws. Firstly, round-trip translation requires extra translation corpus. Secondly, the generated paraphrases are similar to the original sentences in surface structure, as faithful round-trip translation and denoising tend to maintain the syntax, sentence pattern, and tense of the original sentence. Thirdly, most existing methods cannot explicitly control the similarity of the generated paraphrases to the original sentences. Explicitly controlling the similarity allows for greater variations in surface structure and the generation of diverse paraphrases meets the requirements of different downstream tasks. Lastly, the existing methods all ignore the importance of generating accurate entities, while entity errors usually lead to serious factual errors (e.g. paraphrasing "Jobs created Apple" to "Gates created Apple").

To produce paraphrases with great variation in surface structure without translation data, we propose a self-supervised pseudo-data construction method. Also, to enable explicit control of similarity and accurate generation of entities, we propose a paraphrasing model which unifies discrete and continuous variables to encode sentences.[1]

The pseudo-data constructor is trained on non-parallel sentences in a self-supervised manner, eliminating the need for translation data. From

---
* Corresponding author.

[1] The discrete and continuous variables here are both dense representations in neural networks.

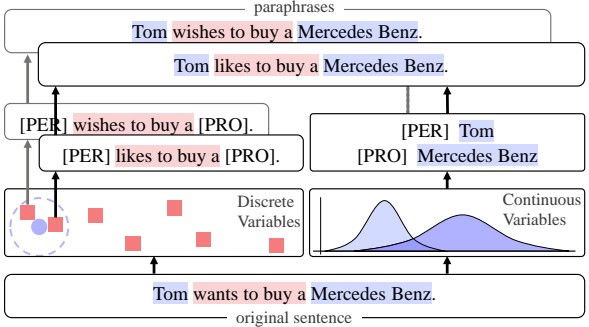

Figure 1: The paraphrasing model unifies discrete and continuous variables to represent the complete semantics. The "[PER]" and "[PRO]" mean the category "person" and "product" of the entities, respectively.

the candidates generated by the constructor, semantically similar and structurally dissimilar paraphrases are selected to train the paraphrasing model, resulting in greater variations in surface structure. The paraphrasing model is inspired by the ideas of applying VQ-VAE to unsupervised paraphrasing (Roy and Grangier, 2019) and the separate modeling of the sentence meaning and the entities (Gu et al., 2016). The model employs discrete variables to encode the sentence meaning and continuous variables to model the entities. As an example, the sentence "Tom wants to buy a Mercedes Benz" could be split into the meaning "someone wants to buy a product" and the entities "someone: Tom, product: Mercedes Benz" as illustrated in Figure 1. The similarity can be controlled by sampling discrete variables at different distances from the original representation. The entities are kept substantially accurate with the specific modeling using continuous variables.

We evaluate our method on two benchmark datasets, WikiAnswers (Fader et al., 2013) and QQP[2]. The automatic evaluations indicate that our pseudo-data construction method surpasses round-trip translation and encourages more varied surface structures. The automatic and human evaluations demonstrate that our paraphrase generation method outperforms the state-of-the-art unsupervised methods on semantic preservation, surface structure variation, and entity retention.

Besides, commonly used metrics such as BLEU (Papineni et al., 2002) and Self-BLEU do not perform a fair evaluation of entity accuracy. To more effectively evaluate the accuracy of generated entities, we introduce a metric called Entity-

---

[2]https://www.kaggle.com/datasets/quora/question-pairs-dataset.

Score, which performs an evaluation of the precision and recall of the entities and correlates significantly better with human evaluation than the existing metrics on the entity.

## 2 Related Works

Unsupervised paraphrase generation methods often employ translation or denoising. Mallinson et al. (2017) and Guo et al. (2021) employ a straightforward round-trip translation approach. Guo et al. (2021) propose a hybrid decoder taking the hidden states of a set-to-sequence model (Vinyals et al., 2016) and a round-trip translation model. Hu et al. (2019a) and Hu et al. (2019b) expand translation datasets into paraphrase datasets with lexically constrained decoding (Hokamp and Liu, 2017; Post and Vilar, 2018). Principled Paraphrasing introduces an adversarial reconstruction decoder to encourage the removal of structural information from the original sentence, followed by a translation task to ensure the retention of semantics (Ormazabal et al., 2022). These methods rely on translation datasets and often produce paraphrases that resemble the original sentences. As for denoising, Liu et al. (2020) model paraphrase generation as an optimization problem and design a sophisticated objective function to enhance semantic preservation, expression diversity, and fluency of paraphrases, failing to explicitly control the similarity to generate diverse paraphrases. Niu et al. (2021) propose Dynamic Blocking to enforce a surface structure dissimilar from the input, which easily leads to key phrases being blocked. Roy and Grangier (2019) propose to learn paraphrasing models from unlabeled monolingual corpus with VQ-VAE (van den Oord et al., 2017). Yet discrete variables are expressively less powerful than continuous variables (Liu et al., 2021), and it is difficult to encode large quantities of entities with a limited number of discrete variables in a codebook.

We present a self-supervised Pseudo-data Constructor that generates paraphrase-like data for training the paraphrasing model, which eliminates the need for the translation corpus. And we propose a paraphrasing model to generate diverse paraphrases with controlled similarity through sampling discrete variables and generate accurate entities through specific modeling entities with continuous variables.

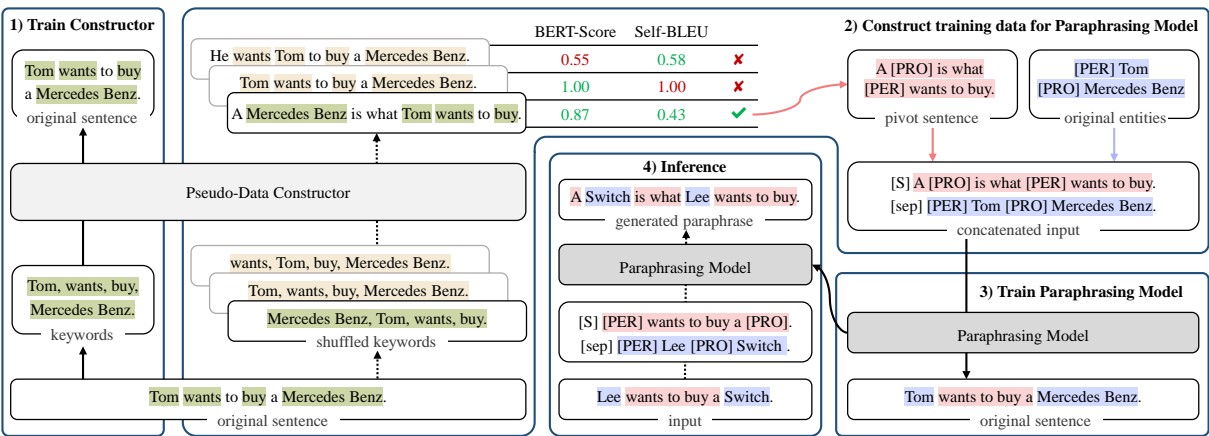

Figure 2: Overview of our approach. The dashed lines with arrow indicate inferencing.

## 3 Approach

We describe the proposed approach in this section. Besides, there are some efforts that show similarities to our work, we give detailed comparisons between them and our method in Appendix A.

### 3.1 Overall Training Procedure

The overview of our approach is depicted in Figure 2. The training is conducted in three steps.

**1) Train pseudo-data constructor.** Traditional pseudo-paraphrase construction methods which rely on round-trip translation introduce two limitations: the need for the translation corpus and domain gaps between the translation corpus and the paraphrase corpus. To remove these limitations, we train a pseudo-data constructor in a self-supervised manner on the non-parallel paraphrase data. As shown in step 1 of Figure 2, we remove the stop words from a sentence and split the remaining words into words and noun chunks as the input keywords. The constructor is trained by reconstructing the original sentence with ordered input keywords.

**2) Construct training data for paraphrasing model.** As depicted in step 2 of Figure 2, we generate paraphrase candidates with varied word orders using the constructor and filter the candidates for semantically similar but structurally dissimilar pseudo-paraphrases to train the paraphrasing model. This encourages variations in word order while preserving semantics. Specifically, we shuffle the keywords of a sentence multiple times and feed them into the constructor. The constructor outputs several candidates corresponding to the orders of the keywords, as the order of the keywords in the target sentence is consistent with the

input within the training. We filter the candidates using BERTScore (Zhang et al., 2020) and Self-BLEU because not all keywords orders yield valid paraphrases. Then each selected candidate and the original sentence form an input-target pair.

To separate modeling the sentence meaning and entities for explicitly controlling the similarity without destroying entities, we further split the meaning and the entities of the input sentence. We replace the entities in the input sentence with the entity categories (e.g. replace "Mercedes Benz" with "[PRO]") to obtain the entity-replaced pivot sentence and concatenate several special tokens, the pivot sentence, a special separator, and the correct entities from the target sentence to serve as the input to the paraphrasing model.[3]

**3) Train paraphrasing model.** The paraphrasing model is able to produce paraphrases with lexical variations and accurate entities. The model utilizes discrete variables to encode sentence meanings and stores these variables in a codebook. The close variables in the codebook tend to express different words with similar semantics and we sample the discrete variables to bring lexical variation. The entities are specifically modeled with continuous variables, which ensures the accuracy of the generated entities. During training, the paraphrasing model takes the concatenated sequence constructed in step 2 as input and outputs the original sentence. We train the paraphrasing model in a standard sequence-to-sequence manner.

---

[3]In this study, we use Spacy (https://spacy.io, v3.4, "en_core_web_trf" model) as the NER tool.

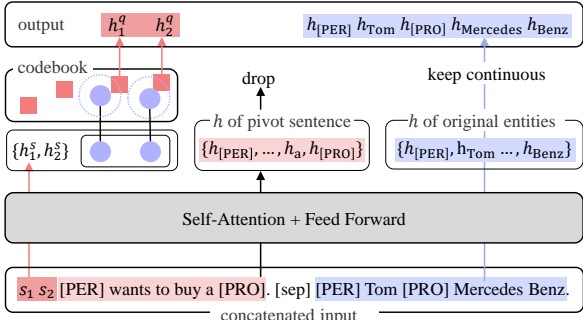

Figure 3: Encoder architecture of the paraphrasing model. "$s_i$" denotes a special token to be quantized, in this figure we take two special tokens as an example.

## 3.2 Paraphrasing Model

Given an original sentence $x = \{w_1, w_2, ..., w_t\}$ and its paraphrase $y$ generated by the pseudo-data constructor, we construct the input as described in step 2 in Section 3.1. Following Roy and Grangier (2019), we quantize the representation of special tokens added at the beginning of the input to encode the sentence meaning. We denote the special tokens with $s = \{s_1, ..., s_m\}$, the entity-replaced pivot sentence with $y^-$, the special separator with [sep], and the entities in $x$ with $a^x = \{a_1^x, ..., a_n^x\}$. The input to the paraphrasing model would be: $\hat{x} = \{s; y^-; [sep]; a^x\}$.

The paraphrasing model is an encoder-decoder architecture that features a codebook containing $K$ vectors. As shown in Figure 3, the encoder is a transformer encoder with a quantization process that converts continuous variables into discrete variables. The input $\hat{x}$ is fed into N-layer self-attention blocks, and the last block outputs the hidden states $h = \{h_1^s, ..., h_m^s, h^{y^-}, h^{[sep]}, h^{a^x}\}$ of each token. The output $h^s = \{h_1^s, ..., h_m^s\}$ of the special tokens are quantized as:

$$h_j^q = c_k, \text{where } k = \underset{i}{\arg\min} \|h_j^s - c_i\|^2,$$

where $c_i$ denotes the $i$-th vector in the codebook. The hidden states of the pivot sentence $y^-$ are dropped, while the hidden states $h^{a^x} = \{h_1^{a^x}, ..., h_n^{a^x}\}$ of the entities $a^x$ are retained and output as continuous variables without being quantized. We concatenate the discrete variables and the continuous variables, i.e. $h^{enc} = \{h_1^q, ..., h_m^q, h_1^{a^x}, ..., h_n^{a^x}\}$, as the final output (also referred to as memory) of the encoder.

A transformer decoder serves as our decoder, which takes $h^{enc}$ as input and outputs the original

sentence $x$. The cross attention to the encoder output allows the model to automatically learn when to focus on discrete variables that represent the sentence meaning and when to focus on continuous variables that represent the entities.

## 3.3 Inference Step

In inference, given a sentence $x$, the entities $a^x$ are recognized from $x$ using a NER tool and replaced with entity categories to form the pivot sentence $x^-$. The special tokens, the pivot sentence, the separator token, and the correct entities sequence are concatenated, similar to the training step 2, to form the input $\hat{x} = \{s; x^-; [sep]; a^x\}$. Different from training, each $h_j^s$ is quantized to $h_j^q$ by sampling a vector from the codebook according to the probability:

$$p(h_j^q) = \text{Softmax}(-\frac{d}{\tau}),$$

where $d = \{d_1, d_2, ..., d_K\}$ means the distances of $h_j^s$ to each vector $c_i$ in the codebook, $d_i = \|h_j^s - c_i\|^2$, and $\tau$ is the sampling temperature that we set manually to control the similarity of the generated and original sentences. As the temperature increases, it becomes more likely to sample discrete variables that are farther away from $h^s$, leading to a greater dissimilarity. Finally, the discrete and continuous variables are fed into the decoder to generate the paraphrase autoregressively.

## 4 Experiments

### 4.1 Datasets

We evaluate our approach on two benchmark datasets, WikiAnswers (Fader et al., 2013) and QQP[4]. The WikiAnswers dataset contains 18,000k question-paraphrase pairs scraped from the WikiAnswers website. The QQP dataset comprises over 400k lines of potential question duplicate pairs. We adopt the unsupervised setting as Guo et al. (2021). For WikiAnswers, we randomly select 500k non-parallel sentences, 3k parallel paraphrases, and 20k parallel paraphrases for training, validation, and testing. For QQP, we randomly select 400k non-parallel sentences, 3k parallel paraphrases, and 20k parallel paraphrases for training, validation, and testing. The statistics of the processed datasets are in Appendix B. In the

---

[4]https://www.kaggle.com/datasets/quora/question-pairs-dataset.

pseudo-data constructing step, we pick the candidates with Self-BLEU below 60 and BERTScore above 85 as pseudo data.

## 4.2 Baselines

We compare our method with four unsupervised paraphrase generation methods, including round-trip translation, Set2seq+RTT (Guo et al., 2021), TA+SS+DB (Niu et al., 2021), and Principled Paraphrasing (Ormazabal et al., 2022). While some of the baseline methods employ parallel translation corpus, we consider them as unsupervised baselines for comparison since they do not rely on parallel paraphrase data.

Set2seq+RTT trains a set2seq (Vinyals et al., 2016) model and a pair of round-trip translation models separately. In generating, a hybrid decoder takes a weighted sum of the hidden states from the set2seq model and the translation model as the probability of the next token. TA+SS+DB applies Dynamic-Blocking to generate paraphrases with no identical n-grams to the input. We note that we prevent the blocking of entities in our replication to generate more accurate entities. Principled Paraphrasing introduces an additional adversarial reconstruction decoder to remove as much structural information from the original sentence as possible. Subsequently, a traditional translation loss is employed to preserve the semantic information. This method allows for the control of similarity to the original sentence by adjusting the weights of the adversarial loss and translation loss during training. These aforementioned methods represent the current state-of-the-art in unsupervised paraphrase generation. By comparing with these approaches, we aim to demonstrate the superiority of our approach in preserving semantics and entities while altering surface structures.

For reference, we also present the results of two supervised methods, BART+finetune (Lewis et al., 2020) and DNPG (Li et al., 2019), and a large language model, GPT-3.5-turbo[5], in Appendix D.

## 4.3 Training Details

For the pseudo-data constructor, we finetune a BART (Lewis et al., 2020) model from Huggingface's checkpoint[6] on the non-parallel training set. The paraphrasing model contains 267M parameters, including a codebook that contains 50M pa-

rameters. We use the same pre-trained BART-base checkpoint as in the constructor to initialize the paraphrasing model except for the codebook. The codebook contains 32,768 vectors of dimension 768. To initialize these vectors, we train the paraphrasing model in a completely continuous style, skipping the quantization, for 4 epochs. Then we feed partial training data into the encoder and cluster the hidden states of the special token into 32,768 classes using k-means to assign values to the vectors in the codebook. In quantization, the hidden states of special tokens are quantized separately with the same codebook. Following van den Oord et al. (2017), the model is trained with straight-through gradient estimation and exponentiated moving averages. The paraphrasing models are trained on the corresponding datasets with 10 epochs on a single 32G Tesla V100 GPU for about 60 minutes per epoch.

## 4.4 Evaluation Metrics

### 4.4.1 Automatic Evaluation

Recent works on paraphrase generation typically take BLEU (Papineni et al., 2002), Self-BLEU, and iBLEU (Sun and Zhou, 2012) as evaluation metrics. Niu et al. (2021) propose BERT-iBLEU which leverages BERTScore and Self-BLEU to measure the semantics and surface structure. Following their works, We take BLEU, Self-BLEU, iBLEU, and BERT-iBLEU[7] as the metrics.

Existing evaluation metrics fail to perform a fair evaluation of entities. To address this limitation, we propose Entity-Score, an F1-score-like evaluation method that performs a holistic evaluation of the precision and recall of the generated entities. Entity-Score calculates precision by determining whether the entity in the paraphrase appears in the input, and recall by assessing whether the entity in the input is present in the paraphrase. The final Entity-Score is then computed as the F1 score based on precision and recall. A more detailed description of the Entity-Score is in Appendix C. We adopt Entity-Score as an evaluation metric.

### 4.4.2 Human Evaluation

We randomly select 50 examples containing entities from QQP, and three annotators are asked to rate the paraphrases generated by our method

---

[5]https://platform.openai.com/docs/models/gpt-3-5.

[6]https://huggingface.co/facebook/bart-base.

[7]Follow Niu et al. (2021), we use "roberta-large_L17_no-idf_version=0.3.0(hug_trans=2.3.0)" to calculate BERTScore.

| Method | BLEU↑ | Self-BLEU↓ | iBLEU↑ | BERT-iBLEU↑ | Entity-Score↑ |
|---|---|---|---|---|---|
| round-trip translation | 27.85 | 55.66 | 19.50 | 75.49 | 49.67 |
| TA+SS+DB | 30.56 | 47.29 | 22.78 | 81.65 | 67.83 |
| Set2seq+RTT | 33.82 | 48.76 | 25.56 | 80.46 | 69.91 |
| Principled Paraphrasing$^\alpha$ | 33.69 | 55.53 | 24.77 | 77.31 | 56.04 |
| Principled Paraphrasing$^\beta$ | 31.25 | 44.18 | 23.71 | 82.01 | 49.00 |
| Ours($\tau$=5) | **38.57** | 48.52 | **29.86** | 81.16 | **85.10** |
| Ours($\tau$=11) | 34.17 | 35.78 | 27.18 | 85.32 | 81.27 |
| Ours($\tau$=12) | 31.59 | **32.11** | 25.22 | **86.15** | 80.79 |

Table 1: Evaluation results on WikiAnswers dataset. $\tau$ means the sampling temperature in quantization. Principled Paraphrasing$^\alpha$ uses $\lambda = 0.85$ and $K = 0.70$. Principled Paraphrasing$^\beta$ uses $\lambda = 0.80$ and $K = 0.70$. And $\lambda$ and $K$ are hyper-parameters in Ormazabal et al. (2022). The best and second best results are indicated in **bold** and underlined, respectively.

| Method | BLEU↑ | Self-BLEU↓ | iBLEU↑ | BERT-iBLEU↑ | Entity-Score↑ |
|---|---|---|---|---|---|
| round-trip translation | 23.25 | 68.12 | 14.11 | 67.56 | 80.76 |
| TA+SS+DB | 14.35 | 36.19 | 9.30 | 84.83 | 74.80 |
| Set2seq+RTT | 22.41 | 51.78 | 14.99 | 80.44 | 74.17 |
| Principled Paraphrasing$^\alpha$ | 24.53 | 73.39 | 14.74 | 63.39 | 55.04 |
| Principled Paraphrasing$^\beta$ | 15.60 | 35.81 | 10.46 | 85.78 | 34.18 |
| Ours($\tau$=5) | **25.58** | 62.78 | **16.74** | 73.51 | **95.35** |
| Ours($\tau$=10) | 23.80 | 57.82 | 15.64 | 76.80 | 94.58 |
| Ours($\tau$=18) | 14.22 | **33.37** | 9.46 | **86.44** | 91.44 |

Table 2: Evaluation results on QQP dataset. Principled Paraphrasing$^\alpha$ uses $\lambda = 0.90$ and $K = 0.70$. Principled Paraphrasing$^\beta$ uses $\lambda = 0.73$ and $K = 0.70$.

($\tau = 5$ and $\tau = 10$) and three strong baselines according to the semantics, structure, and entity on a scale of 1 to 3, the higher, the better. The semantics measures the degree of preservation of semantics. The structure evaluates the extent of variation in surface structure. The entity measures the accuracy and completeness of the generated entities. The details of the human evaluation settings are given in Appendix F.

### 4.5 Evaluation Results

#### 4.5.1 Automatic Evaluation Results

We select the models that perform best on the dev set and report their performances on the test set. Table 1 and Table 2 present the results on WikiAnswers and QQP. Several generated examples are provided in Appendix H. The following conclusions can be drawn from the results:

**1) Our approach outperforms the state-of-the-art unsupervised methods on existing metrics at appropriate temperatures.** To ensure a fair and unbiased comparison, we carefully select sampling temperatures that generate paraphrases with comparable BLEU scores to the baseline methods. The results presented in Table 1 indi-

cate that our proposed method exhibits superior performance in BLEU, iBLEU, and Entity-Score when $\tau$=11. Furthermore, when $\tau$=12, our method outperforms the baseline methods in Self-BLEU, BERT-iBLEU, and Entity-Score. Similar trends can be observed in the results displayed in Table 2.

**2) Our approach showcases exceptional proficiency in generating accurate entities.** Our method consistently achieves a significantly higher Entity-Score compared to other methods, and the Entity-Score remains high as the temperature rises, which demonstrates the effectiveness of our method in preserving entities.

**3) The similarity between the generated paraphrases and the original sentences in surface structure can be explicitly controlled by adjusting the sampling temperature during quantization.** At low sampling temperatures, our method maintains good semantics, scoring high in BLEU and Self-BLEU. While at high sampling temperatures, our method generates paraphrases with greater variation in surface structure and scores well in Self-BLEU. The detailed evaluation results on different sampling temperatures can be found in Appendix E.

| | Method | BLEU↑ | Self-BLEU↓ | iBLEU↑ | BERT-iBLEU↑ | Entity-Score↑ |
|---|---|---|---|---|---|---|
| WikiAnswers | Round-trip Translation | 27.85 | 55.66 | 19.50 | 75.49 | 49.67 |
| | Pseudo-data Constructor | **32.08** | **28.20** | **26.05** | **88.84** | **77.12** |
| QQP | Round-trip Translation | **23.25** | 68.12 | **14.11** | 67.56 | 80.76 |
| | Pseudo-data Constructor | 17.35 | **34.11** | 12.20 | **88.12** | **85.25** |

Table 3: Experimental results of different pseudo-data construction methods.

| | Semantic | Structure | Entity |
|---|---|---|---|
| Set2seq+RTT | 2.37 | 2.03 | 2.67 |
| TA+SS+DB | 2.11 | 2.26 | 2.52 |
| Principled$^{\alpha}$ | 2.30 | 1.89 | 2.06 |
| Principled$^{\beta}$ | 1.95 | 2.12 | 1.75 |
| Ours($\tau$=5) | **2.77** | 2.10 | **2.97** |
| Ours($\tau$=10) | 2.55 | **2.41** | 2.89 |
| Agreement | 0.45 | 0.46 | 0.87 |

Table 4: The results of human evaluation. "Principled" means Principled Paraphrasing. Principled$^{\alpha}$ uses $\lambda = 0.90$ and $K = 0.70$. Principled$^{\beta}$ uses $\lambda = 0.73$ and $K = 0.70$. We use Fleiss' Kappa (Fleiss, 1971) to measure the inter-rater agreement. The scores between 0.40-0.60 mean "moderate agreement", and the scores above 0.80 mean "very good agreement".

| Metric | Correlation Value |
|---|---|
| BLEU | 0.257 ($p < 0.01$) |
| Self-BLEU | 0.256 ($p < 0.01$) |
| iBLEU | 0.185 ($p < 0.01$) |
| BERT-iBLEU | -0.152 ($p < 0.01$) |
| Entity-Score | 0.579 ($p < 0.01$) |

Table 5: Spearman's correlation between the automatic evaluation and the human evaluation score on entity.

### 4.5.2 Human Evaluation Results

Table 4 shows the results of human evaluation on QQP. Our method with $\tau$=10 outperforms other baselines in three metrics and our method with $\tau$=5 shows significant superiority over other baselines in semantics and entities. As the sampling temperature rises, the variation in the paraphrases generated by our method increases. Furthermore, our method obtains significantly higher scores in entity, which highlights the superiority of our method in generating accurate entities. The results of the human evaluation are compatible with the results of the automatic evaluation.

### 4.6 Analyses

#### 4.6.1 Pseudo-data Constructor

To compare the performance of our Pseudo-data Constructor and Round-trip Translation in constructing pseudo-data, we apply the Pseudo-data Constructor and Round-trip Translation directly on the test sets of WikiAnswers and QQP datasets. For both the Constructor and Round-trip translation[8], we generate 5 candidates for each sentence and then select the best one as the pseudo-paraphrase based on BERT-iBLEU.[9] We employ nucleus sampling (Holtzman et al., 2020) with $p$=0.95 to generate multiple candidates in translation because the translation models do not feature diverse generation.

As shown in Table 3, the Pseudo-data Constructor outperforms Round-trip Translation in most metrics. Particularly, the Pseudo-data Constructor far outperforms Round-trip Translation on Self-BLEU, which demonstrates that the Pseudo-data Constructor is more likely to generate pseudo-data with greater variation in surface structure. Notably, the accuracy of the entities generated by Round-trip Translation on the two different datasets varies substantially, and our constructor generates entities with higher accuracy on both datasets. This might be related to the domain of the training data for the translation models.

#### 4.6.2 Effectiveness of Entity-Score

To verify the effectiveness of Entity-Score, we compute the Spearman's correlation between the automatic evaluation scores and the human evaluation score of the entity. As shown in Table 5, Entity-Score correlates clearly better with human evaluation than existing metrics in evaluating generated entities. More details about the correlation are given in Appendix G.

---

[8]We use the translation models on Huggingface's checkpoint (https://huggingface.co/facebook/wmt19-en-de and https://huggingface.co/facebook/wmt19-de-en).

[9]BERT-iBLEU is a composite metric calculated by BERTScore and Self-BLEU, which makes the filtering criteria here coincide with Section 3.1.

| Method | BLEU↑ | Self-BLEU↓ | iBLEU↑ | BERT-iBLEU↑ | Entity-Score↑ |
|---|---|---|---|---|---|
| round-trip translation | 23.25 | 68.12 | 14.11 | 67.56 | 80.76 |
| +LCD | $23.39_{(+0.14)}$ | $71.06_{(-2.94)}$ | $13.95_{(-0.16)}$ | $64.85_{(-2.71)}$ | $83.82_{(+3.06)}$ |
| Principled Paraphrasing | 24.53 | 73.39 | 14.74 | 63.39 | 55.04 |
| +LCD | $\underline{25.18}_{(+0.65)}$ | $76.24_{(-2.85)}$ | $15.04_{(+0.3)}$ | $60.13_{(-3.26)}$ | $72.92_{(+17.88)}$ |
| Ours (Discrete) | 24.12 | **59.01** | 15.81 | **74.62** | 60.76 |
| +LCD | $18.78_{(-5.34)}$ | $\underline{60.35}_{(-1.34)}$ | $10.87_{(-4.94)}$ | $73.04_{(-1.58)}$ | $\underline{86.57}_{(+25.81)}$ |
| +Continuous | $\mathbf{25.58}_{(+1.46)}$ | $62.78_{(-3.77)}$ | $\mathbf{16.74}_{(+0.93)}$ | $\underline{73.51}_{(-1.11)}$ | $\mathbf{95.35}_{(+34.59)}$ |

Table 6: Results of methods combined with lexically constrained decoding (LCD). Discrete in our method means that only discrete variables are used in inference. The results in brackets indicate the variation in performance after combining the LCD and continuous variables, with "+" indicating an improvement and "-" indicating a decline.

| Model | BLEU↑ | Self-BLEU↓ | iBLEU↑ | BERT-iBLEU↑ | Entity-Score↑ |
|---|---|---|---|---|---|
| Ours ($\tau$=5) | 25.58 | 62.78 | 16.74 | 73.51 | 95.35 |
| w/o pseudo data | 23.36 | 68.17 | 14.21 | 68.95 | 94.98 |
| w/o entity | 23.85 | 57.45 | 15.72 | 74.94 | 59.55 |
| w/o continuous | 24.12 | 59.01 | 15.81 | 74.62 | 60.67 |

Table 7: Evaluation results on ablation study. The sampling temperature is set to 5. Without pseudo data means we train the paraphrasing model in a denoising manner without constructing pseudo data. Without entity means that we do not feed the entities to the encoder and the decoder only takes the quantized hidden states of special tokens. Without continuous means that we quantize the hidden states of special tokens and entities.

### 4.6.3 Comparison with Lexically Constrained Decoding

Our paraphrasing model encodes entities individually using continuous variables to improve the accuracy of the generated entities. Lexically constrained decoding (LCD) on entities offers the potential to achieve this as well. On the QQP dataset, we analyze the impact of LCD on our method (only using discrete variables) and two baselines that obtain near BLEU scores with our method, to demonstrate the superiority of continuous encoding compared to LCD. Table 6 shows the experimental results. It can be seen that LCD improves the entity accuracy on all methods, but our method still far outperforms the others. In our approach, the use of continuous variables is more effective than LCD, and LCD causes a reduction in the other metrics. The results demonstrate the validity of our use of continuous variables to encode entities individually.

### 4.6.4 Ablation Study

To better observe what roles the different modules play in our approach, we remove one module at a time from our approach and train paraphrasing models on the non-parallel training set. Table 7 shows the performance of our approach on QQP after removing different modules. Pseudo-data constructor increases the variation in surface structure as "w/o pseudo data" score higher in Self-BLEU. Taking entities as part of the input to the paraphrasing model and keeping the hidden states of entities as continuous variables are crucial to generate accurate entities, since "w/o entity" and "w/o continuous" drop a lot in Entity-Score. In our approach, various modules play critical roles, cooperating to generate semantically similar, surface-structurally distinct paraphrases that contain accurate and complete entities in an unsupervised setting.

## 5 Conclusion

In this paper, we propose a self-supervised pseudo-data construction method that generates paraphrases with more variation in surface structure than round-trip translation without the need for translation data. We propose a paraphrasing model that employs discrete and continuous variables to represent sentence meaning and entity information respectively for unsupervised paraphrase generation. Our approach achieves state-of-the-art performance on two benchmark paraphrase datasets and far outperforms other methods in terms of accuracy in generating entities. In addition, we proposed an evaluation metric called *Entity-Score* for evaluating the precision and recall

of the generated entities. Entity-Score correlates better with the human evaluation of entity than existing metrics. To the best of our knowledge, this is the first work concerned with ensuring the quality of the generated entities in the paraphrase generation task. The framework encoding text with discrete and continuous variables is promising on many NLP tasks, e.g. summarization and translation. Future work will focus on extending our model to a more general seq2seq framework.

## Limitations

To the best of our knowledge, our approach has the following two limitations. First, the paraphrasing model has great advantages when dealing with sentences containing entities but will reduce to the traditional VQ-VAE when dealing with sentences that do not contain entities. Second, our approach employs an additional named entity recognition tool. The performance of the named entity recognition tool we use will directly affect the performance of our approach.

Regarding the Entity-Score, there are two limitations that we are currently aware of. First, like in our approach, the process of calculating the Entity-Score requires the use of a named entity recognition tool. Thus, the Entity-Score results are directly influenced by the tool. Second, in the paraphrase generation tasks, some entities can exist in different forms, such as "$2,000" and "two thousand dollars". Currently, we do not optimize the Entity-Score calculation process for these kinds of cases.

In terms of experiments, we only conduct experiments on two English question datasets, WikiAnswers and QQP. We choose these two datasets because 1) they are two widely used benchmark datasets, 2) they contain a large number of entities to validate the performance of our proposed paraphrasing method and Entity-Score, and 3) we follow the experimental settings with Niu et al. (2021).

Regarding the choice of pre-trained model, we only experiment on BART. We focus on the problem of accurate entity and controlled surface structure similarity in paraphrasing tasks, with the key point being the coordination of discrete and continuous variables. We choose BART simply because it is a commonly used seq2seq pre-trained model. We believe that using other pre-trained models, such as T5 (Raffel et al., 2020), would not affect our experimental conclusions.

## Ethics Statement

We declare that all of the co-authors of this paper are aware of and honor the ACL Code of Ethics. In this paper, we propose an effective pseudo-data construction method, a paraphrasing model, and an evaluation metric on entities. Our contributions lie in the methodological aspect and do not bring any ethical issues. However, The publicly released datasets and pre-trained models used in this paper may contain some negative effects, such as gender and religious bias. We recommend other researchers stay careful when using our methods with these datasets and models.

We conduct a human evaluation to rate the generated results, and in doing so, we ensure that no personal information of any of the annotators is involved in this work. We reviewed the samples to be rated before the human evaluation to make sure there are no negative statements. All the annotators were informed in advance of the potential risks of offensive statements generated by artificial intelligence.

## Acknowledgments

This work is supported by the Fundamental Research Funds for the Central Universities under Grant 1082204112364 and the Key Program of the National Science Foundation of China under Grant 61836006.

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

# A Methodological Comparisons

## A.1 Wordset for Paraphrasing

Our proposed Pseudo-data Constructor uses keywords as input and outputs the original sentence during training. In inference, we use the shuffled keywords as input which prompt the constructor to generate sentences that match the order of the keywords, which makes it possible for each sentence to correspond to multiple possible paraphrases to enrich the diversity of pseudo data. We additionally use BERT-iBLEU to filter data pairs with large semantic offsets or too similar structures to ensure the quality of the pseudo-data.

There are some works that use the change of the wordset orders to paraphrase. Fu et al. (2019) first predict the neighbors of the words in the original sentence to form the bag of words then sample and reorganize the words into sentences. Huang and Chang (2021) and Guo et al. (2021) encode the input sentence using a transformer encoder without positional encoding to ignore its surface structure. Reconstructing the original sentence using unordered keywords reduces the diversity of paraphrases because the same wordset always yields the same output regardless of the order of the words. Therefore, we train the constructor by preserving the order of keywords and inference by shuffling the keywords to obtain diverse paraphrase pseudo-data. Goyal and Durrett (2020) guide the decoder to output syntactically meeting sentences by reordering the syntax of the input and appending the new syntactic position information to the encoder output. This approach requires complex rearrangement and evaluation of the syntax tree and uses it as additional input. Therefore, we choose a simpler pseudo-data construction process of shuffling and filtering.

## A.2 Discrete and Continuous Variables

Our paraphrasing model uses discrete and continuous variables to encode sentence meanings and entities, respectively. Hosking and Lapata (2021); Hosking et al. (2022) use discrete and continuous variables to encode the syntax and semantics of the sentences. In terms of the model structure, both of their models and our paraphrasing model consist of an encoder that outputs discrete and continuous variables and a decoder that accepts discrete and continuous variables. The differences are a) their methods are supervised, b) their methods require additional templates with the same syntax as the

label as input, c) they train another syntax predictor to identify the target syntax while we sample the presentations with different surface structures from discrete codebook, and d) they aim to generate paraphrases of different syntaxes while we emphasize the accuracy of entities in generating paraphrases of different surface structures.

## A.3 Lexically Constrained Decoding

We use continuous variables in the paraphrasing model to improve the accuracy of the entities and lexically constrained decoding (Hokamp and Liu, 2017) has the potential to accomplish this task. The difference is that our approach meets this constraint on the input and model side with a *soft* training objective, while lexically constrained decoding implements this on the decoding side with a *hard* occurrence constraint. This hard constraint leads to a degradation of the generation quality because the constraint that some tokens must be generated results in the sampling of suboptimal text during the generation process. We compare our approach with the lexically constrained decoding and demonstrate the superiority of our approach in Section 4.6.3.

## A.4 Explicit Control of Similarity

We adopt discrete variable sampling to obtain multiple paraphrases of the same sentence, and we can control the similarity between the paraphrase and the original sentence in terms of surface structure by adjusting the sampling temperature. Some decoding strategies, such as top-k sampling (Fan et al., 2018), nucleus sampling (Holtzman et al., 2020), and DIPS (Kumar et al., 2019) support similar functionality. The difference is that our discrete variables encode the meaning and structure of the whole sentence, and sampling on discrete variables implies sampling at the sentence level, while the sampling of the above decoding strategies goes at the token level. Further, our sampling is performed on the model side (input side for decoder) and the sampling of decoding strategies is performed on the decoding side, which also means that our method can be used simultaneously with these decoding strategies without conflicts.

The existing works that can control the similarity by hyperparameters also include VQ-VAE (Roy and Grangier, 2019) and Principled Paraphrasing (Ormazabal et al., 2022). Our approach draws on the discrete sampling method of VQ-VAE, with the difference that a) VQ-VAE em-

ploys discrete variables to encode sentences while we enhance the accuracy of entities with continuous variables and b) VQ-VAE is trained using a reconstruction objective while we use an additional Pseudo-data Constructor to further enhance the diversity of paraphrases. Principled Paraphrasing uses an additional adversarial reconstruction decoder to encourage the encoder to remove as much structural information from the original sentence as possible, and then employs a translation task to motivate the encoder to retain the semantics. This approach can control the similarity of the paraphrases by controlling the loss weights of reconstruction and translation. Besides the need for additional parallel translation data, Principled Paraphrasing requires setting the loss weights before training, which means that it requires training multiple models with different hyperparameters to generate paraphrases with different degrees of similarity. In contrast, our approach controls the similarity by sampling discrete variables in the inference phase, in other words, one trained model can be used to generate diverse paraphrases.

## B The Statistics of the Datasets

Table 8 shows the statistics of the processed datasets.

|      | Train / Val / Test | Avg token | Avg entity |
|------|--------------------|-----------|------------|
| Wiki | 500k / 3k / 20k    | 8.25      | 0.26       |
| QQP  | 400k / 3k / 20k    | 11.33     | 0.44       |

Table 8: Statistics of WikiAnswers and QQP. The training sets contain non-parallel sentences and the val/test sets contain parallel paraphrase-pairs. "Avg token" means the average number of tokens per sentence and "avg entity" means the average number of entities per sentence.

## C Details of Entity-Score

We use $N$ to denote the number of input sentences, $a$ to denote an entity, $A_n^i$, $A_n^r$, and $A_n^g$ to denote the set of entities in the n-th input sentence, in the references of the n-th input sentence, and in the generated paraphrase of the n-th input sentence.

In calculating the precision, we consider an entity as true positive if the entity in the generated paraphrase appears in the input sentence or reference sentences, otherwise false positive, i.e.

| Task | Method | BLEU↑ | Self-BLEU↓ | iBLEU↑ | BERT-iBLEU↑ | Entity-Score↑ |
|------|--------|-------|------------|--------|-------------|---------------|
| WikiAnswers | bart+finetune | 37.20 | 37.40 | 29.74 | 85.52 | 70.10 |
| | DNPG | **41.64** | 33.26 | **34.15** | - | - |
| | GPT-3.5-turbo | 20.93 | **15.75** | 17.26 | **91.26** | 66.71 |
| | Ours($\tau$=5) | 38.57 | 48.52 | 29.86 | 81.16 | **85.10** |
| QQP | bart+finetune | **31.85** | 37.98 | **24.87** | 86.29 | 68.72 |
| | DNPG | 25.03 | 45.17 | 18.01 | - | - |
| | GPT-3.5-turbo | 9.30 | **15.77** | 6.79 | **92.46** | 72.78 |
| | Ours($\tau$=5) | 25.58 | 62.78 | 16.74 | 73.51 | **95.35** |

Table 9: Evaluation results on supervised methods and large language model.

$$\text{TP} = \sum_{n=1}^{N} \sum_{a \in A_n^g} f_{\text{TP}}(a); \ f_{\text{TP}}(a) = \begin{cases} 1, & a \in (A_n^i \cup A_n^r) \\ 0, & otherwise \end{cases},$$

$$\text{FP} = \sum_{n=1}^{N} \sum_{a \in A_n^g} f_{\text{FP}}(a); \ f_{\text{FP}}(a) = \begin{cases} 0, & a \in (A_n^i \cup A_n^r) \\ 1, & otherwise \end{cases},$$

$$\text{precision} = \frac{\text{TP}}{\text{TP} + \text{FP}}.$$

In calculating the recall, we consider an entity as false negative if the entity in the input sentence does not appear in the generated paraphrase, i.e.

$$\text{FN} = \sum_{n=1}^{N} \sum_{a \in A_n^i} f_{\text{FN}}(a); \ f_{\text{FN}}(a) = \begin{cases} 1, & a \notin A_n^g \\ 0, & otherwise \end{cases},$$

$$\text{recall} = \frac{\text{TP}}{\text{TP} + \text{FN}}.$$

Thereafter, we calculated entity-score with precision and recall like F1-Score, i.e.

$$\text{entity-score} = \frac{2 \times \text{precision} \times \text{recall}}{\text{precision} + \text{recall}}.$$

## D  Results on Supervised Methods and GPT-3.5-turbo

When experimenting with GPT-3.5-turbo, we use a simple instruction "Paraphrase the following sentence and keep the entities correct:". Table 9 shows the results on supervised methods and GPT-3.5-turbo.

It can be seen that there are still gaps between our method and the supervised methods, especially on the surface structure, as our method scores much higher in Self-BLEU. Surprisingly, our approach achieves the highest Entity-Scores on both datasets, even in the face of a powerful large language model. It is worth noting that although GPT-3.5-turbo performs poorly in BLEU, we analyze some of the generated examples and

find that the paraphrasing quality of GPT-3.5-turbo is much better than other methods in subjective ratings. We assume that this is because the labels of the paraphrase datasets are still quite inadequate in terms of diversity and surface structure variation. Also, GPT-3.5-turbo tends to generate paraphrases that are longer than the original sentences, which directly leads to poor performance on BLEU scores.

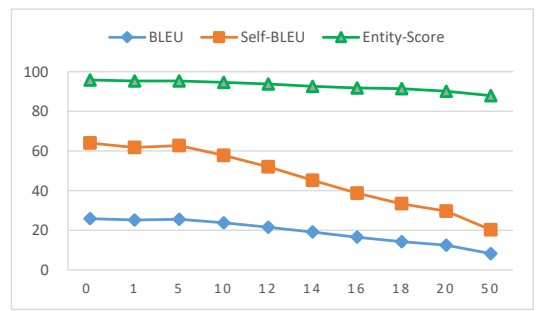

Figure 4: BLEU, Self-BLEU and entity-score on QQP change with increasing sampling temperature.

| $\tau$ | BLEU | SB | iB | Bert-iB | ES |
|--------|------|-----|-----|---------|-----|
| $1^{-9}$ | 25.87 | 63.99 | 16.88 | 72.62 | 95.82 |
| 5 | 25.58 | 62.78 | 16.74 | 73.51 | 95.35 |
| 10 | 23.80 | 57.82 | 15.64 | 76.80 | 94.58 |
| 12 | 21.58 | 52.06 | 14.22 | 79.97 | 93.80 |
| 14 | 19.14 | 45.17 | 12.71 | 82.97 | 92.60 |
| 16 | 16.55 | 38.77 | 11.02 | 85.09 | 91.74 |
| 18 | 14.22 | 33.37 | 9.46 | 86.44 | 91.44 |
| 20 | 12.49 | 29.70 | 8.27 | 87.12 | 90.10 |
| 50 | 8.25 | 20.25 | 5.40 | 88.20 | 87.93 |

Table 10: The experimental results with rising sampling temperatures. For a friendly look, we use SB, iB, Bert-iB, and ES to denote Self-BLEU, iBLEU, Bert-iBLEU, and Entity-Score, respectively. $\tau$=$1^{-9}$ means that we disable the sampling.

In comparison with supervised methods, our approach does not require parallel paraphrase data.

Compared to large language models, our approach consumes much fewer computational resources during training and inference.

## E  Evaluation Results on Different Sampling Temperatures

Figure 4 shows the changes in BLEU, Self-BLEU, and Entity-Score on the QQP dataset with rising sampling temperatures in our method. Table 10 presents the detailed experimental results. As the temperature rises, the Entity-Score still maintains decent although both BLEU and Self-BLEU drop substantially, which demonstrates that we can control the similarity to a large extent while maintaining the accuracy of the generated entities.

We observed the generated paraphrases at a high ($\tau = 50$) sampling temperature. There is a part of the paraphrases that deviates significantly from the original sentences in semantics although these paraphrases achieve a high average Bert-iBLEU score. This is the reason why we select sampling temperatures that yield paraphrases with similar BLEU to the baselines to compare in Table 1 and Table 2.

## F  Human Evaluation Details

We randomly select 50 examples containing entities from QQP. Three annotators are asked to rate the paraphrases generated by our method ($\tau = 5$ and $\tau = 10$) and three strong baselines according to the semantics, surface structure, and entity on a scale of 1 to 3, the higher the better. The annotators rate the generated paraphrases as bad/normal/good, corresponding to 1/2/3 points respectively. Before the rating, we informed all the annotators of the potential risks caused by the negative statements generated by artificial intelligence. As shown in Figure 5, Figure 6, and Figure 7, we present detailed scoring guidelines and examples rated with different scores. The annotators were recruited in the authors' labs and they all have relevant English publications in the field of neural networks. Payment for this human evaluation is made in full by the lab's supervisor based on the workload.

We use Fleiss' Kappa (Fleiss, 1971) to measure the inter-rater agreement and the results are 0.45 (moderate agreement), 0.46 (moderate agreement), and 0.87 (very good agreement) in semantics, structure, and entity. The agreement score demonstrates the consistency among the three annotators.

## G  Correlation Between Automatic and Human Evaluation

Table 11 presents the Spearman's correlation between the automatic evaluation metrics and the human evaluation scores. Entity-Score correlates better with human evaluation on entity than other automatic metrics. Self-BLEU correlates well on semantics, surface structure, and entity. This is because more word overlap tends to imply higher semantic similarity, less structural variation, and a greater possibility of containing the same entities. BERT-iBLEU shows the opposite result to Self-BLEU. This may be caused by Self-BLEU being over-considered in calculating BERT-iBLEU. Unexpectedly, BLEU correlates poorly with human evaluation of semantics. This is due to the fact that paraphrase generation is a highly open-ended task. And iBLEU does not seem to be a good evaluation metric either, because iBLEU uses BLEU as the key to evaluate the semantics.

It is not fair enough to use a single metric (iBLEU or BERT-iBLEU) to evaluate the quality of the paraphrases as in previous works. Our method outperforms unsupervised baselines in multiple automatic metrics and human evaluation that demonstrates the superiority of our method.

## H  Generated Examples

Table 12 shows several examples generated using our method and the unsupervised baselines. Our approach generates paraphrases with greater variations in surface structure, which include more word order variations (sample 3-6), lexical variations (sample 3-6), and introduced extensions (sample 1 and 6). At different sampling temperatures, our method generates paraphrases with different surface structural similarities to the original text, while the semantics are well preserved. When the input contains entities, our method generates more accurate entities than the other baselines (sample 1-5).

| | Semantics | Surface Structure | Entity |
|---|---|---|---|
| BLEU | 0.174 ($p < 0.01$) | -0.124 ($p = 0.03$) | 0.257 ($p < 0.01$) |
| Self-BLEU | 0.401 ($p < 0.01$) | -0.386 ($p < 0.01$) | 0.256 ($p < 0.01$) |
| iBLEU | 0.055 ($p = 0.34$) | -0.012 ($p = 0.83$) | 0.185 ($p < 0.01$) |
| BERT-iBLEU | -0.326 ($p < 0.01$) | 0.377 ($p < 0.01$) | -0.152 ($p < 0.01$) |
| Entity-Score | 0.278 ($p < 0.01$) | -0.027 ($p = 0.65$) | 0.579 ($p < 0.01$) |

Table 11: Spearman's correlation between the automatic evaluation metrics and the human evaluation scores.

## Human Evaluation on Semantics

We are glad that you help us with the human evaluation. One thing to note is that **the sentences to be rated are generated by artificial intelligence**. Although we have done a preliminary review of these sentences, they may still contain offensive information. **If you find any sentences with offensive statements (racial bias, gender bias, religious bias, etc.), please contact us and we will immediately replace the samples at issue.**

Paraphrase Generation aims to generate a sentence that has the same semantic meaning as the input but with a different sentence structure (grammar, sentence pattern, and words). In this evaluation, we need you to rate the paraphrases according to **whether it contains the same semantics as the input.**

- If the sentence and the input contain different semantics, please select [bad].
- If the sentence and the input contain similar semantics, please select [normal].
- If the sentence and the input contain the exact same semantics, please select [good].

Here are some examples:
- input: what do you think about this movie?
- s1: do you like Jackie Chen? [bad]
- s2: do you like this movie? [normal]
- s3: what is your opinion about this movie? [good]
- s4: what do you think of the movie? [good]

---

\* 1. on an average, how much rank or marks are required to get selected in indian telecommunication service through ies?

| | bad | normal | good |
|---|---|---|---|
| how much rank and marks are required to get selected in indian ies through telecommunication service on an average? | ○ | ○ | ○ |

Figure 5: The scoring page for human evaluation on the semantics.

# Human Evaluation on Structure

We are glad that you help us with the human evaluation. One thing to note is that **the sentences to be rated are generated by artificial intelligence**. Although we have done a preliminary review of these sentences, they may still contain offensive information. **If you find any sentences with offensive statements (racial bias, gender bias, religious bias, etc.), please contact us and we will immediately replace the samples at issue.**

Paraphrase Generation aims to generate a sentence that has the same semantic meaning as the input but with a different sentence structure (grammar, sentence pattern, and words). In this evaluation, we need you to rate the paraphrases according to **how much if differs from the input in the surface structure.**

- If the sentence and the input are the same, please select [bad].
- If the sentence and the input differ in just words but similar in grammar, please select [normal].
- If the sentence and the input differ greatly in grammar and pattern, please select [good].

Here are some examples:
- input: what do you think about this movie?
- s1: do you like Jackie Chen? [good]
- s2: do you like this movie? [good]
- s3: what is your opinion about this movie? [normal]
- s4: what do you think about this movie? [bad]

---

\* 1. on an average, how much rank or marks are required to get selected in indian telecommunication service through ies?

|  | bad | normal | good |
|---|---|---|---|
| how much rank and marks are required to get selected in indian ies through telecommunication service on an average? | ◯ | ◯ | ◯ |

Figure 6: The scoring page for human evaluation on the surface structure.

# Human Evaluation on Entity

We are glad that you help us with the human evaluation. One thing to note is that **the sentences to be rated are generated by artificial intelligence**. Although we have done a preliminary review of these sentences, they may still contain offensive information. **If you find any sentences with offensive statements (racial bias, gender bias, religious bias, etc.), please contact us and we will immediately replace the samples at issue.**

Paraphrase Generation aims to generate a sentence that has the same semantic meaning as the input but with a different sentence structure (grammar, sentence pattern, and words). In this evaluation, we need you to rate the paraphrases according to **whether the sentence contain accurate entities as input**.

- If the sentence contains no entities or wrong entities, please select [bad].
- If the sentence contains a part of the entities in the input, please select [normal].
- If the sentence contains all the entities in the input, please select [good].

Here are some examples:
- input: what do you think about the movie Titanic and Leonardo DiCaprio?
- s1: do you like Jackie Chen? [bad]
- s2: do you like this movie? [bad]
- s3: what is your opinion about Titanic? [normal]
- s4: what do you think about Leonardo DiCaprio and Titanic? [good]

---

\* 1. on an average, how much rank or marks are required to get selected in indian telecommunication service through ies?

|  | bad | normal | good |
|---|---|---|---|
| how much rank and marks are required to get selected in indian ies through telecommunication service on an average? | ○ | ○ | ○ |

Figure 7: The scoring page for human evaluation on the entity.

| | |
|---|---|
| Original | what are some possible solutions if i forgot my `icloud` password? |
| Set2seq+RTT | what are the possible solutions if i forget my password? |
| TA+SS+DB | what are the possible solutions if i forget my `iphone` `icloud` password? |
| Principled$^\alpha$ | what are some possible solutions if I forgot my `i[fr] Quelle` password? |
| Principled$^\beta$ | what solutions can be found if I have lost my `imple` password? |
| Ours($\tau = 5$) | i forgot my `icloud` password, what are some solutions? |
| Ours($\tau = 10$) | i forgot my `icloud` password. what are the solutions to this? |
| Original | will the `nintendo switch` become successful? |
| Set2seq+RTT | is the `ncero switch` successful? |
| TA+SS+DB | will `nintendo's the switch` become successful? |
| Principled$^\alpha$ | will the `nintendo switch` become successful? |
| Principled$^\beta$ | will the `nintendo switch` be successful? |
| Ours($\tau = 5$) | will the `nintendo switch` ever become successful? |
| Ours($\tau = 10$) | will `nintendo switch` ever become successful? |
| Original | which is the best book with which i can prepare for `gre exams` ..? |
| Set2seq+RTT | what are the best books to prepare for the `greco examination` ? |
| TA+SS+DB | which book is best for the `gre` ? with which books can i prepare for exams.. |
| Principled$^\alpha$ | which is the best book with which i can prepare for `gre exams` ? |
| Principled$^\beta$ | what's the best book for me to prepare for my exams? |
| Ours($\tau = 5$) | how do i prepare for `gre exams` ..which is the best book for it? |
| Ours($\tau = 10$) | which is the best book for preparation of `gre exams` ..? |
| Original | was the `suicide squad` movie good in your opinion? |
| Set2seq+RTT | do you think the film of the `suicide squad` is good? |
| TA+SS+DB | was `suicide squad` a good movie in your opinion? |
| Principled$^\alpha$ | was the `suicide squad` movie good in your opinion? |
| Principled$^\beta$ | do you think the movie about the `suicidequad` is a good one? |
| Ours($\tau = 5$) | is the `suicide squad` movie any good in your opinion? |
| Ours($\tau = 10$) | what is your opinion on the movie `suicide squad` ? |
| Original | how many days does it take a `pan card` to arrive after applying? |
| Set2seq+RTT | how many days after applying for a `panc card` ? |
| TA+SS+DB | how many days does a `pan card` take to arrived in `india` after apply? |
| Principled$^\alpha$ | how many days does it take a `pan card` to arrive after applying? |
| Principled$^\beta$ | how many days does it take to arrive after the application to receive a card? |
| Ours($\tau = 5$) | when applying for `pan card` , how many days does it take to arrive? |
| Ours($\tau = 10$) | when applying for `pan card` , how many days does it take for the `pan card` to arrive? |
| Original | how should i stop being insecure? |
| Set2seq+RTT | how do i stop being insecure? |
| TA+SS+DB | how should i stopped being so insecure? |
| Principled$^\alpha$ | how should i stop being insecure? |
| Principled$^\beta$ | how can i stop being insecure? |
| Ours($\tau = 5$) | how can i stop being insecure? |
| Ours($\tau = 10$) | what can i do to stop being insecure about myself? |

Table 12: The examples generated using our method and two baselines. The entities in original sentences are indicated in blue . The correct/wrong entities in generated paraphrases are indicated in green / red .