# OpenReview forum: "Unifying Discrete and Continuous Representations for Unsupervised Paraphrase Generation"
_EMNLP/2023/Conference — EMNLP 2023 Main_

### Official Review · Reviewer_Lw6x · 2023-07-31

**Soundness:** 3

**Excitement:**

3: Ambivalent: It has merits (e.g., it reports state-of-the-art results, the idea is nice), but there are key weaknesses (e.g., it describes incremental work), and it can significantly benefit from another round of revision. However, I won't object to accepting it if my co-reviewers champion it.

**Paper Topic And Main Contributions:**

This paper proposes an approach for unsupervised paraphrase generation through unifying discrete and continuous representation. Unsupervised paraphrase generation is important due to the labor intensive effort in annotation in the supervised setting. Existing approaches in unsupervised paraphrase generation rely on back-translation or denoising, which suffer from multiple limitations - a) back-translation results in additional corpus of another language, b) the generated paraphrases of the original unsupervised approaches are very similar to the original sentences, c) existing unsupervised approaches could not control the syntactical similarity of the generated paraphrases and d) the generation of correct entities in the paraphrases are ignored. This paper is intended to tackle these limitations. This paper proposes a self-supervised pseudo-data construction approach (trained on non-parallel corpora) and a paraphrasing model that unifies both discrete and continuous variables (discrete variables are used for encoding the sentence meaning and the continuous variables are used to model the entities) to encode sentences. The datasets used are WikiAnswers and Quora. Results show that the pseudo-data construction approach has more varied syntactical paraphrases than back-translation and the overall experimental results are promising. The paper also introduces a new metric, Entity-Score, to evaluate the precision and recall of the entities.

**Reasons To Accept:**

- Novel approach of unsupervised paraphrase generation.
- The results showed that the proposed approach is promising and performs better than existing state-of-the-art unsupervised paraphrase generation approaches.
- Propose a novel metric, Entity-Score, to evaluate the precision and recall of the entities.


**Reasons To Reject:**

- Some parts of the approaches are unclear. It is unclear how the pseudo-data constructor is trained with the shuffled keywords. In the paper, it's just one small paragraph that is used to describe this process. It is also unclear if the generated sentences (the candidates) are grammatically incorrect, since the word shuffling may result in grammatical incorrect sentences.
- Entities need to be created during inferencing. It is unclear if the entities can be output from the same model.
- The existing back-translation dataset e.g., ParaNMT, which is a very popular unsupervised paraphrasing dataset, should be evaluated to see how the proposed approach really performs with it. ChatGPT, should also be used to compare the performance, since it is also some-what of an unsupervised setting

**Reproducibility:**

3: Could reproduce the results with some difficulty. The settings of parameters are underspecified or subjectively determined; the training/evaluation data are not widely available.

**Reviewer Confidence:**

4: Quite sure. I tried to check the important points carefully. It's unlikely, though conceivable, that I missed something that should affect my ratings.

---

> ### Author Rebuttal · Authors · 2023-08-25
>
> Thank you for taking the time to review the paper. We apologize if some parts of the manuscript were not comprehensible enough and led to misinterpretations. Hopefully, our response will help to alleviate any concerns you may have.
>
> $\text{\color{blue}Reason 1: It is unclear how the pseudo-data constructor is trained with the shuffled keywords.}$
>
> Answer 1: As stated in lines 181-183, "the constructor is trained by reconstructing the original sentence with ordered input keywords". We do not shuffle the keywords in training constructor. This information is further illustrated in step 1 of Figure 2.
>
> $\text{\color{blue}Reason 2: It is unclear if the generated sentences (the candidates) are grammatically incorrect.}$
>
> Answer 2: As stated in lines 197-200, "we filter the candidates using BERTScore (Zhang et al., 2020) and Self-BLEU because not all keywords orders yield valid paraphrases". We have acknowledged the potential grammatical incorrectness and have included a screening step specifically aimed at identifying and mitigating this problem. We also visualize this phenomenon and the corresponding screening process in step 2 of Figure 2.
>
> $\text{\color{blue}Reason 3: It is unclear if the entities can be output from the same model.}$
>
> Answer 3: The entities can not be output from our paraphrasing model. Specifically, we employ Spacy to recognize the entities, as duly stated in line 212. You raised an interesting concern, as we mentioned in the Limitations (lines 569-573), the reliance on an external tool can be considered as one of the limitations of our proposed method. However, this does not affect the effectiveness of our method and the experimental conclusions.
>
> $\text{\color{blue}Reason 4: The existing back-translation dataset e.g., ParaNMT, should be evaluated.}$
>
> Answer 4:  Following Niu et al.(2021), we demonstrate the effectiveness of our method on two commonly used datasets, WikiAnswers and QQP. We also explain the reasons for our choice of datasets in lines 585-593. ParaNMT and other back-translation datasets do not exhibit properties that are not available in WikiAnswers and QQP and are relevant to the goals of our approach. So we believe that our method will achieve promising results on these datasets as well. We will try to add the ParaNMT results in the modified version.
>
> $\text{\color{blue}Reason 5: ChatGPT should also be used to compare the performance.}$
>
> Answer 5: We provide experimental results and analysis of ChatGPT in Appendix D, with notes in the Section Baselines in lines 343-346. However, we must consider that ChatGPT is a large-scale language model, and we have limited information about its training data and architecture, even we cannot be sure whether there is any test data leakage. Therefore, it would be unfair to compare our method directly with ChatGPT. Nonetheless, we include the experimental results in the appendix as a reference and provide corresponding analysis.

---

### Official Review · Reviewer_PrBd · 2023-08-03

**Soundness:** 5

**Excitement:**

4: Strong: This paper deepens the understanding of some phenomenon or lowers the barriers to an existing research direction.

**Missing References:**

- L473 should cite Holtzman et al. (2020)


**Paper Topic And Main Contributions:**

This paper describes a method for unsupervised paraphrasing that does not rely on translation corpora and is capable of producing diverse paraphrases.
Unlike past work on unsupervised paraphrasing, the method is explicitly focused on maintaining the faithfulness of entity mentions.
The authors propose a VQ-VAE architecture trained on data extracted by pseudo-labelling.
The central idea of the VQ-VAE approach is that the sentence syntax can be handled in a discrete and quantized space, allowing the authors to sample high variance sentences, while the entity mentions are handled separately.
By separating the entities from the rest of the sentence, the method allows for greater diversity while still maintaining entity faithfulness.

The authors evaluate their method using automatic and human evaluation.
In addition to standard automatic metrics, the authors introduce a new metric which focuses specifically on entity F1.
The proposed method outperforms past work and is competitive with supervised methods in some cases.
The summary of contributions is
1. a VQ-VAE method for paraphrasing which maintains entity mentions
2. a pseudo-labelling method for creating paraphrase data
3. a metric for scoring entity mention faithfulness

**Questions For The Authors:**

- How does the method handle cases where entities are close in distributional space? I'm thinking of a case like "President of the USA" and "Barack Obama", which might be very close in distributional space since they co-occur frequently, but where one cannot replace "President" with "Obama" in all cases.
- Why did you choose BART instead of T5? It seems to me like the setting in Fig. 3 bears a strong resemblance to the T5 pre-training/inference regime and the use of extra tokens to bind variables. Have you tried your method with T5?
- Maybe I'm misunderstanding the baselines, but have you tried running the pseudo-data constructor on its own as a paraphrase model? How much better is the existing model (especially at entities) than the constructor alone?
- what was the basis for choosing the number of vectors in the codebook?

**Reasons To Accept:**

1. Experimental results. The results of the method are strong and meet the goals set out in the introduction. The authors also include relevant ablations. The human evaluation makes me more confident in the quality of the results.

2. Relevant baselines. The authors compare against relevant baselines. Especially relevant is the comparison/inclusion of lexically-constrained decoding, which can be combined with their method.

3. Well-written and presented. For the most part, I was able to understand the paper immediately. The model and analysis is presented in a generally well-organized and well-written way. The related work section is well-written and situates the method well. I especially liked the presentation of differences in Table 5.

**Reasons To Reject:**

1. Extension of past work. As far as I can see, the model here is an improvement on/extension of the VQ-VAE from Roy and ABC (2019).
However, this is not a major weakness, as the authors make targeted modifications for entity faithfulness and also use pre-trained models, which makes the method more relevant to current NLP efforts.



**Reproducibility:**

4: Could mostly reproduce the results, but there may be some variation because of sample variance or minor variations in their interpretation of the protocol or method.

**Reviewer Confidence:**

4: Quite sure. I tried to check the important points carefully. It's unlikely, though conceivable, that I missed something that should affect my ratings.

**Typos Grammar Style And Presentation Improvements:**

- I really don't like the notation $y^{-e}$, $e^{x}$, etc.. I was immediately confused as $e$ already has a common use in equations, especially exponentials like $e^x$.

---

> ### Author Rebuttal · Authors · 2023-08-25
>
> We are excited that you like our work and now we offer the following responses to your questions.
>
> $\text{\color{blue}Q1: How does the method handle close entities like "President" and "Obama"?}$
>
> A1: We encode entities using continuous variables, and we do not apply sampling and replacement on continuous variables during training and inference, so the close entities, like "President" and "Obama", will not be intentionally replaced.
>
> $\text{\color{blue}Q2: Why did you choose BART instead of T5?}$
>
> A2: We focus on the problem of accurate entity and controlled surface structure similarity in paraphrasing tasks, with the key point being the coordination of discrete and continuous variables. We choose BART simply because it is a commonly used seq2seq pre-trained model. We believe that using other pre-trained models, such as T5, would not affect our experimental conclusions.
>
> $\text{\color{blue}Q3: How about running the pseudo-data constructor on its own as a paraphrase model?}$
>
> A3: We do not treat the constructor as a baseline and consider it as one of the essential modules of our method. We recognize that the constructor can be used on its own as a paraphrasing model. So we compare the constructor with another pseudo-data construction method, round-trip translation, in 4.6.1. If we directly compare our method with the constructor, our method significantly outperforms the constructor in syntactic preservation and entity accuracy.
>
> $\text{\color{blue}Q4: What was the basis for choosing the number of vectors in the codebook?}$
>
> A4: There is no consensus on the size of the codebook from previous works (Roy and Grangier, 2019; Hosking and Lapata, 2021), we explore multiple sizes and subsequently identify the most optimal one based on its performance.
>
> In addition, we will add the reference and optimize the notation as suggested.

---

### Official Review · Reviewer_PEo6 · 2023-08-05

**Soundness:** 5

**Excitement:**

5: Transformative: This paper is likely to change its subfield or computational linguistics broadly. It should be considered for a best paper award. This paper changes the current understanding of some phenomenon, shows a widely held practice to be erroneous in someway, enables a promising direction of research for a (broad or narrow) topic, or creates an exciting new technique.

**Paper Topic And Main Contributions:**

-

**Reasons To Accept:**

-

**Reasons To Reject:**

-

**Reproducibility:**

5: Could easily reproduce the results.

**Reviewer Confidence:**

1: Not my area, or paper was hard for me to understand. My evaluation is just an educated guess.

---

> ### Author Rebuttal · Authors · 2023-08-28
>
> Thank you for your positive comments about our work!

---

### Meta-Review · Area_Chair_z1Lw · 2023-09-20

**Recommendation:** 4

**Metareview:**

The paper presents a unique approach to unsupervised paraphrase generation by blending discrete and continuous representation. This method stands out from earlier unsupervised techniques, aiming to produce diverse syntactical paraphrases without the need for translation corpora while maintaining entity accuracy. At its heart, the methodology utilizes a VQ-VAE architecture combined with a self-supervised pseudo-data construction approach trained on non-parallel corpora. For assessment purposes, the research employed the WikiAnswers and Quora datasets and introduced the Entity-Score metric to measure the precision and recall of entities.
Regarding the topic and contributions, both reviewers concur that the paper presents a fresh perspective on unsupervised paraphrase generation, with a particular emphasis on maintaining entity integrity. They both underscore the innovative use of a VQ-VAE architecture and acknowledge the introduction of the Entity-Score metric as a significant advancement.
In terms of strengths, there's a unanimous sentiment about the promising nature of the experimental results. Both reviewers laud the novel approach and express appreciation for the Entity-Score metric, particularly highlighting the thorough evaluation bolstered by human assessments.
On the flip side, clarity appears to be an issue. The reviewers both identify areas in the paper, such as the methodology of the pseudo-data constructor and the treatment of entities, that could benefit from more explicit elaboration. They recommend additional evaluations using renowned datasets or models to further validate the approach.
Additionally, while Reviewer 3 praises the clarity and structure of the paper, Reviewer 2, though not explicitly critical, refrains from complimenting its writing style.
Overall, despite recognizing the paper's strengths and contributions, the reviewers have different areas of focus and depth in their critiques.

NB.: The assessment based on the two reviews should suffice for now. Decisions can still be made from those reviews, or if deemed necessary, another reviewer could be approached to provide further insights at a later time.

---

### Decision · Program_Chairs · 2023-10-07

**Decision:**

Accept-Main

**Comment:**

The paper presents a unique approach to unsupervised paraphrase generation by blending discrete and continuous representation. This method stands out from earlier unsupervised techniques, aiming to produce diverse syntactical paraphrases without the need for translation corpora while maintaining entity accuracy. At its heart, the methodology utilizes a VQ-VAE architecture combined with a self-supervised pseudo-data construction approach trained on non-parallel corpora. For assessment purposes, the research employed the WikiAnswers and Quora datasets and introduced the Entity-Score metric to measure the precision and recall of entities.
Regarding the topic and contributions, both reviewers concur that the paper presents a fresh perspective on unsupervised paraphrase generation, with a particular emphasis on maintaining entity integrity. They both underscore the innovative use of a VQ-VAE architecture and acknowledge the introduction of the Entity-Score metric as a significant advancement.
In terms of strengths, there's a unanimous sentiment about the promising nature of the experimental results. Both reviewers laud the novel approach and express appreciation for the Entity-Score metric, particularly highlighting the thorough evaluation bolstered by human assessments.
On the flip side, clarity appears to be an issue. The reviewers both identify areas in the paper, such as the methodology of the pseudo-data constructor and the treatment of entities, that could benefit from more explicit elaboration. They recommend additional evaluations using renowned datasets or models to further validate the approach.
Additionally, while Reviewer 3 praises the clarity and structure of the paper, Reviewer 2, though not explicitly critical, refrains from complimenting its writing style.
Overall, despite recognizing the paper's strengths and contributions, the reviewers have different areas of focus and depth in their critiques.

NB.: The assessment based on the two reviews should suffice for now. Decisions can still be made from those reviews, or if deemed necessary, another reviewer could be approached to provide further insights at a later time.